SCIENCE FORUM

# Considerations when investigating lncRNA function in vivo

**Abstract** Although a small number of the vast array of animal long non-coding RNAs (lncRNAs) have known effects on cellular processes examined in vitro, the extent of their contributions to normal cell processes throughout development, differentiation and disease for the most part remains less clear. Phenotypes arising from deletion of an entire genomic locus cannot be unequivocally attributed either to the loss of the lncRNA per se or to the associated loss of other overlapping DNA regulatory elements. The distinction between *cis*- or *trans*-effects is also often problematic. We discuss the advantages and challenges associated with the current techniques for studying the in vivo function of lncRNAs in the light of different models of lncRNA molecular mechanism, and reflect on the design of experiments to mutate lncRNA loci. These considerations should assist in the further investigation of these transcriptional products of the genome.

**ANDREW R BASSETT\*, ASIFA AKHTAR, DENISE P BARLOW, ADRIAN P BIRD, NEIL BROCKDORFF, DENIS DUBOULE, ANNE EPHRUSSI, ANNE C FERGUSON-SMITH, THOMAS R GINGERAS, WILFRIED HAERTY, DOUGLAS R HIGGS, ERIC A MISKA AND CHRIS P PONTING\***

**\*For correspondence:**
andrew.bassett@path.ox.ac.uk
(ARB); chris.ponting@dpag.ox.
ac.uk (CPP)

**Reviewing editor**: Detlef Weigel, Max Planck Institute for Developmental Biology, Germany

Complex transcription interwoven between and within protein-coding genes produces many thousands of long non-coding RNAs (lncRNAs) that are greater than 200 nucleotides (nt) in length but that appear to lack protein-coding potential (*Djebali et al., 2012*). Nevertheless, even for the earliest discovered lncRNAs, such as mammalian *H19*, *Xist* or fruitfly *roX*, molecular effects and functional significance have proven difficult to establish (*Gabory et al., 2010*; *Ilik et al., 2013*; *Sado and Brockdorff, 2013*). Furthermore, no or only subtle mouse phenotypes were revealed by detailed loss-of-function studies of *Malat1* or *Evf-2*. In contrast, mutation of *Fendrr* results in early lethality, and targeted replacement of *BC1* results in seizures for some mice (*Table 1*). It is not possible to accurately predict from the level or extent of its expression, or its sequence composition, whether disruption of a lncRNA locus will result in an overt phenotype. This makes loss- or gain-of-function experiments crucial to understanding the roles of lncRNAs in vivo.

Many lncRNAs are known to act as primary host transcripts for classes of small non-coding RNAs (*da Rocha et al., 2008*; *Royo and Cavaille, 2008*). However, lncRNAs are also presumed to regulate the expression either of their neighbouring genes in *cis*, or of more distant genes in *trans* (*Figure 1*). The function of a lncRNA may be mediated by the gene's RNA product which can bind to proteins or to other nucleic acids thereby modulating their functions. This could act by competing with endogenous mRNAs for miRNA binding (*Franco-Zorrilla et al., 2007*; *Poliseno et al., 2010*; *Jeck and Sharpless, 2014*), providing binding sites for small RNAs that elicit transcriptional silencing (*Wierzbicki et al., 2009*), or through altering protein activity (*Feng et al., 2006*), binding or specificity (reviewed in *Guttman and Rinn, 2012*). Alternatively, the act of transcription per se through a lncRNA locus could be critical because of the changes this generates in chromatin structure, modification or protein binding: in this case the resultant RNA could be an incidental by-product

**Table 1.** Representative studies that have disrupted lncRNA loci in vivo (N/A—not applicable)

| lncRNA name | Organism | Mutation strategy | Reported animal phenotype | RNA-based rescue? | Reference |
|---|---|---|---|---|---|
| *Xist* | *Mus musculus* | ~15 kb replaced with a *neo* expression cassette | Females inheriting paternal allele were embryonic lethal; males fully viable | No | (*Marahrens et al., 1997*) |
| *Xist* | *Mus musculus* | Inversion of Exon 1 to intron 5 | Embryonic lethality of paternally inherited allele | No | (*Senner et al., 2011*) |
| *H19* | *Mus musculus* | Replacement by *neo* cassette | Slightly increased growth | No | (*Ripoche et al., 1997*) |
| *roX* | *Drosophila melanogaster* | Deletions of *roX1* or *roX2* | None, except when in combination: male-specific reduction in viability | Yes | (*Meller and Rattner, 2002*) |
| *Kcnq1ot1* | *Mus musculus* | Promoter deletion | Growth deficiency for paternally inherited mutation | No | (*Fitzpatrick et al., 2002*) |
| *Airn* | *Mus musculus* | Premature transcriptional termination | Growth deficiency for paternally inherited mutation | No | (*Sleutels et al., 2002*) |
| *Evf2* | *Mus musculus* | Premature transcriptional termination | None | N/A | (*Bond et al., 2009*) |
| *BC1* | *Mus musculus* | Replacement of promoter and exon by *PgkNeo* cassette | Vulnerable to epileptic fits after auditory stimulation | No | (*Zhong et al., 2009*) |
| *Neat1* | *Mus musculus* | 3 kb Promoter and 5' deletion | None | N/A | (*Nakagawa et al., 2011*) |
| *Tsx* | *Mus musculus* | 2 kb Promoter and exon 1 deletion | Smaller testes and less fearful (males) | No | (*Anguera et al., 2011*) |
| *Malat1* | *Mus musculus* | Deletion | None | N/A | (*Eissmann et al., 2012*) |
| *Malat1* | *Mus musculus* | *lacZ* insertion and premature transcriptional termination | None | N/A | (*Nakagawa et al., 2012*) |
| *Malat1* | *Mus musculus* | 3 kb Promoter and 5' deletion | None | N/A | (*Zhang et al., 2012*) |
| *Hotair* | *Mus musculus* | Deletion | Spine and wrist malformations | No | (*Li et al., 2013*) |
| *Hotdog* and *Twin of Hotdog* | *Mus musculus* | Large (28 Mb) translocation by inversion | Loss of *Hoxd* expression in the cecum | N/A | (*Delpretti et al., 2013*) |
| *Fendrr* | *Mus musculus* | Replacement of exon 1 with transcriptional stop signal | Embryonic lethal around E13.75 | Yes (majority of embryos) | (*Grote et al., 2013*) |
| *Fendrr* | *Mus musculus* | Locus replacement with *lacZ* cassette | Perinatal lethality | No | (*Sauvageau et al., 2013*) |
| *Peril* | *Mus musculus* | Locus replacement with *lacZ* cassette | Perinatal lethality | No | (*Sauvageau et al., 2013*) |
| *Mdgt* | *Mus musculus* | Locus replacement with *lacZ* cassette | Reduced viability and reduced growth | No | (*Sauvageau et al., 2013*) |
| *15 other lncRNA loci* | *Mus musculus* | Locus replacement with *lacZ* cassette | None | N/A | (*Sauvageau et al., 2013*) |

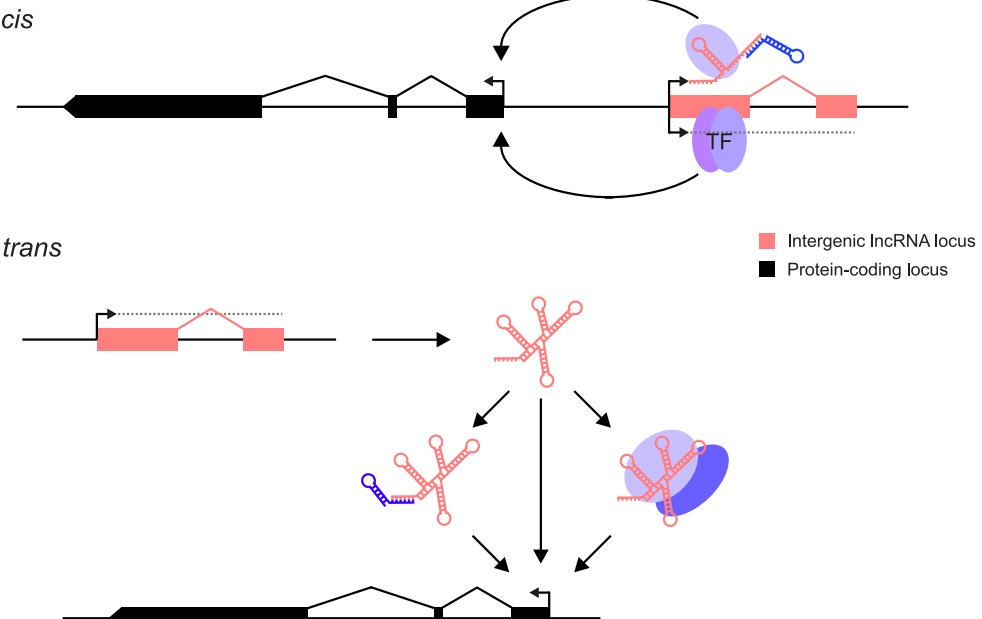

*cis*

*trans*

■ Intergenic lncRNA locus
■ Protein-coding locus

**Figure 1**. lncRNAs can act through *cis* and/or *trans* mechanisms. lncRNAs (pink) can act to regulate expression of their genomically neighbouring protein-coding genes (black) in *cis* (upper panel), or of distant protein-coding genes in *trans* (lower panel). In both situations, the RNA moiety itself may act through binding to cellular proteins (blue ovals) or via base-pairing with other RNAs (blue stem-loop) to modulate their function or binding. The RNA may also directly bind double-stranded DNA in *trans* (**Grote et al., 2013**) or in *cis* (**Senner et al., 2011**). The lncRNA locus (pink) may also encompass transcription factor binding sites (TF) that regulate the transcription of neighbouring genes. This effect may either be entirely independent of the lncRNA, or the binding of transcription factors may be affected positively or negatively by the act of transcription through the lncRNA locus. In this case, the mature RNA product would be incidental.

(**Petruk et al., 2006**; **Latos et al., 2012**; **Marquardt et al., 2014**). In these latter cases, any technique intended to dissect mechanism must alter the act and extent of transcription rather than change RNA levels. This multiplicity of lncRNA functional mechanisms means that a toolkit of experimental strategies to dissect their modes of action will need to be added to those currently employed for investigating protein-coding genes. Protein-coding genes have been shown to contribute greatly to biological function, which is not yet the case for lncRNA loci, rendering their rigorous investigation particularly important.

Analysis of lncRNA localisation both on a tissue and subcellular level by techniques such as fluorescent in situ hybridisation (FISH, **Chakraborty et al., 2012**) can give important insights into the cell types that are important for their function, and in which subcellular compartment they act. Understanding the mechanism of action of lncRNAs often relies on identification of interacting proteins or nucleic acids by RNA-protein (e.g., crosslinking immunoprecipitation, CLIP, **Huppertz et al., 2014**), RNA–RNA (e.g., crosslinking analysis

of synthetic hybrids, CLASH, **Helwak et al., 2013**) or RNA-DNA (e.g., CHART, **Simon et al., 2011**; **Vance and Ponting, 2014** and ChIRP, **Chu et al., 2011**) interaction assays. However, due to the nature of the RNA molecule, many assays are prone to non-specific binding, and it is critical to ensure that appropriate controls are performed (**Brockdorff, 2013**). Several of these techniques have therefore been designed to identify direct interactors by crosslinking, and subsequent use of denaturing conditions to remove non-specific interactions. These techniques are clearly important in determining the mechanism of action of lncRNAs, and are critical to guide experimental genetic knockout design. However, an understanding of the functional importance of lncRNAs in the context of the whole organism still relies on manipulating their expression by genetic modification, overexpression or knockdown strategies, and analysis of the resulting phenotypes.

The earliest studied lncRNAs were those associated with imprinting, such as *Airn* and *H19*, or X chromosome regulation, such as *Xist* or *roX1/2* (**Table 1**). In these cases, lncRNA expression was initially linked genetically to a known phenotype,

and cell line models accurately reflected the in vivo models (*Hao et al., 1993*; *Keniry et al., 2012*; *Latos et al., 2012*; *Helwak et al., 2013*; *Huppertz et al., 2014*). These results highlight that the early models of lncRNAs involved in imprinting and X chromosomal dosage compensation could act as paradigms for the study of lncRNAs today (*Kohtz, 2014*). In the absence of a priori phenotypic associations, some lncRNAs have been chosen for study on the basis of their tissue restricted patterns of expression, sequence conservation, or cellular localisation. Others, such as *MALAT1* (whose level of expression is associated with metastasis) have been selected on the basis of their suggested association with disease. *Neat1* and *Malat1* (also known as *Neat2)* are linked loci that produce highly expressed lncRNAs whose sequences are well conserved across diverse mammals and which have specific nuclear localisations (*Chu et al., 2011*). In cells, *Neat1* was shown to be essential for nuclear paraspeckle assembly and maintenance (*Clemson et al., 2009*; *Sasaki and Hirose, 2009*; *Sunwoo et al., 2009*; *Mao et al., 2011a, 2011b*; *Zhang et al., 2012*) and *Malat1*/*Neat2* binds to the Polycomb 2 (PC2) protein which is required for activating growth-control genes (*Vance and Ponting, 2014*). Nevertheless, in vivo disruption of either of these lncRNA loci results in viable and fertile mouse models (*Table 1*).

Confirmation or rejection of lncRNA functionality requires experimental evidence that clearly separates the role of the genomic locus from the role of its RNA products. Here we recommend experimental techniques that achieve this separation whilst minimising disruption of the DNA sequence. Furthermore, we propose some considerations that may assist in interpreting phenotypes arising from mutation of a lncRNA or lncRNA locus (*Box 1*).

## In vivo, loss-of-function strategies

Different genetic loss-of-function strategies can be employed in vivo to study the function of lncRNAs (*Figure 2*). Prioritisation of strategy should depend on the lncRNA's known biology, including its localisation to one or more of the cytoplasm, nucleus or chromatin. In one study, the majority of human lncRNAs were enriched in the cytoplasm (*van Heesch et al., 2014*) and these may associate with ribosomes and, contrary to expectations, some may be translated (*Guttman et al., 2013*; *Kim et al., 2014*; *Wilhelm et al., 2014*). Nuclear lncRNAs, particularly those that are chromatin-associated, could act as *cis*-acting

transcriptional regulators, whereas cytoplasmic or nucleoplasmic lncRNAs might be predicted to function in *trans*; by contrast, some nucleoplasmic lncRNAs may of course be non-functional products of transcription.

Depletion of protein-coding transcripts is often achieved using RNAi-based techniques, which supply double-stranded RNA that is able to trigger post-transcriptional destabilisation of the mature mRNA and inhibit translation, predominantly in the cytoplasm. Although the presence of active RNAi factors in human cell nuclei has been proposed (*Gagnon et al., 2014*) the extent to which exclusively nuclear lncRNAs can be knocked down remains unclear. Whilst useful for studies of many *trans*-acting lncRNAs, RNAi-based knockdown acts post-transcriptionally, and therefore does not block the act of transcription, precluding analyses of lncRNAs which may produce their effects via this mechanism.

Another experimental approach is to genetically manipulate the lncRNA locus. When inserting transcriptional terminator sequences care must be taken to control for changes in spacing between DNA regulatory elements and to take account of regulatory elements that may be inadvertently inserted, such as promoters of resistance genes, since these may be able to drive expression of neighbouring genes or divert activities from nearby enhancers. Insertion of exogenous sequences can induce phenotypes (*Steshina et al., 2006*). Even single *loxP* sites can attract germline methylation that might potentially repress flanking regulatory elements (*Rassoulzadegan et al., 2002*). Extra controls are thus needed to identify possible gain-of-function effects arising from inserted sequences, such as reporters or selection cassettes. The advent of programmable nucleases (*Kim and Kim, 2014*) provides opportunities to investigate these possibilities. Transcriptional terminator sequences can vary in their efficacy depending on the genomic context into which they are inserted, which can cause termination to be highly inefficient. For example, a sequence that efficiently terminates transcription in multiple contexts in *Airn*, failed to do so when inserted close to a CpG island (*Latos et al., 2012*).

Other approaches include deletion of the full-length lncRNA locus or its promoter sequence, mutation of putative functional domains or targeted interruption between the promoter and the RNA sequence through an engineered inversion (*Figure 2*; *Table 1*). Whilst useful, such strategies may not always be successful. Promoter inversion, for instance, may not always abrogate transcription, because of the bidirectionality of

# Box 1. Considerations when interpreting phenotypes resulting from lncRNA mutation

## General Considerations

- The design of functional experiments should be guided by the essential RNA biology of the chosen lncRNA locus: its proximity to protein-coding genes, its chromatin signatures, stability, copy number, full-length transcript models and tissue expression profiles. If it shares a bidirectional promoter then minimise interference with the adjacent locus when designing targeting strategies. If more abundant and stable, with promoter-like chromatin marks at its transcriptional start site, then consider whether the lncRNA acts in *trans* in an RNA dependent manner.
- Consider all available transcript, regulatory element and evolutionary evidence when designing mutations.
- Consider whether, contrary to initial expectations, the lncRNA encodes protein or, as for H19, harbours a miRNA.
- Choice of loss-of-function strategy and prediction of whether the lncRNA acts in *cis* or in *trans* should be informed by its cytoplasmic, nuclear or chromatin localisation. If found in the cytoplasm, consider whether it is, in fact, translated. If chromatin-associated consider whether it acts in *cis*. In contrast, if cytoplasmic or nucleoplasmic, consider whether it is *trans*-acting.
- Choose cells for functional experiments in which the lncRNA is relatively highly expressed, certainly at greater than one molecule per cell.
- Minimise genomic sequence disruptions when investigating lncRNA or lncRNA locus function. Use control manipulations to distinguish disruptions influencing flanking genes from those influencing the lncRNA.
- Investigate each locus using multiple complementary strategies, for example introduction of minimal targeted DNA deletions, inversions or disruptions and, separately, of transcriptional truncation cassettes. Consider using controls for genetic manipulations of lncRNA loci: inverting the truncation cassette where possible, using a mutated truncation cassette, using a different type of truncation cassette, and using different sites to truncate the lncRNA. It is important to remove any selection cassettes and to consider the influence of reporter genes and *loxP* sites on the locus. Fully describe the mutated locus, including whether the selection cassette is retained.

- Assay biological replicates separately. Embryonic stem (ES) or induced pluripotent (iPS) cells frequently vary in their differentiation kinetics, especially after undergoing gene targeting and selection, and mouse embryos, particularly early implantation stage mouse embryos, show considerable variation in developmental timing. Similarly, cancer cell lines are inherently genetically unstable. This variability makes it essential to study multiple clones of cells or independently derived mutants to ensure that the effects observed are due to the mutation of interest, and not dependent on other effects of the genetic background. This is especially important when the phenotypic effects are subtle.

## Assessment of evidence for lncRNA functionality

- Consider the evidence for each of the many known transcriptional or post-transcriptional, nuclear or cytoplasmic, *cis* or *trans*, RNA-dependent or -independent mechanisms of lncRNAs.
- Employ RNAi-based techniques principally when investigating cytoplasmic RNAs and post-transcriptional RNA-dependent mechanisms. If using RNAi, the knockdown effect on the cytoplasmic and nuclear compartment should be determined separately. An alternative is to use antisense DNA oligos to induce an RNase H activity in the nucleus.
- Only claim that a phenotype is caused by alteration of a *trans*-acting lncRNA transcript when it is successfully and repeatedly rescued upon expression of the lncRNA from an independent transgene.
- Take advantage of carefully controlled biochemical approaches when assessing the potential function of a lncRNA.

## Publications and reporting

- Assess and report objectively all evidence for or against RNA sequence-dependent function or transcription-dependent (RNA sequence-independent) function.
- Report phenotypes precisely. Commonly, gene knockouts kill embryos at critical periods for example, implantation, gastrulation, 12.5dpc when the cardiovascular system become essential, and at birth when lungs and many other systems become essential. In general the maternal organs rescue many organ defects of the embryo. For ES cells, phenotypes affecting pluripotency need to be defined and should be considered with caution due to the inherent instability of this state.
- Explicitly caution when evidence for RNA-dependent vs–independent function, or *trans*- vs *cis*-acting function, is not clear-cut.

promoters (*Wu and Sharp, 2013*), and promoter deletion may also disrupt the expression level of protein-coding transcripts with which lncRNAs share a bidirectional promoter. In all of these cases, it is important to minimise the removal or reorganisation of regulatory factor binding sites or other regulatory elements within the DNA, and to control for the addition of novel binding sites. For example, it should be borne in mind that many lncRNAs initiate within enhancers

Bassett *et al.* eLife 2014;3:e03058. DOI: 10.7554/eLife.03058

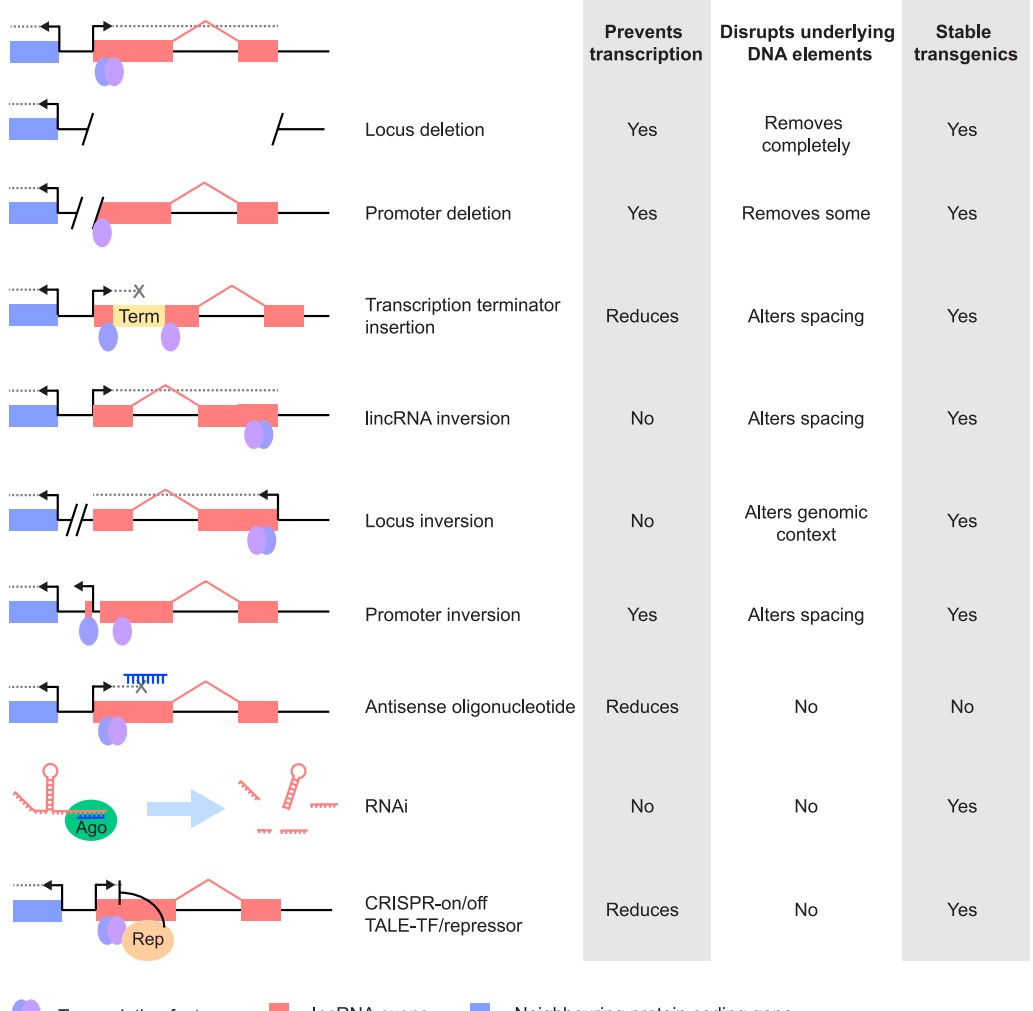

| | Prevents transcription | Disrupts underlying DNA elements | Stable transgenics |
|---|---|---|---|
| Locus deletion | Yes | Removes completely | Yes |
| Promoter deletion | Yes | Removes some | Yes |
| Transcription terminator insertion | Reduces | Alters spacing | Yes |
| lincRNA inversion | No | Alters spacing | Yes |
| Locus inversion | No | Alters genomic context | Yes |
| Promoter inversion | Yes | Alters spacing | Yes |
| Antisense oligonucleotide | Reduces | No | No |
| RNAi | No | No | Yes |
| CRISPR-on/off TALE-TF/repressor | Reduces | No | Yes |

Transcription factors     lncRNA exons     Neighbouring protein coding gene

**Figure 2**. Different strategies for analysis of lncRNA loss-of-function. Strategies that have been used to alter lncRNA function are described pictorially, with the wild type situation on the top-most line. The lncRNA locus is indicated in pink, neighbouring protein-coding gene in blue, transcription factor binding sites within it by blue and purple ovals, transcriptional terminator sequences in yellow ('Term') and the process of transcription by grey dotted lines. Antisense oligonucleotides are able to bind to nascent RNA transcripts and trigger RNase H mediated degradation of the transcript in the nucleus. RNAi is elicited by short RNA species that bind to argonaute proteins (Ago, green oval) within the cell. This complex recognises complementary lncRNA molecules in the cytoplasm, and triggers their destabilisation by the endogenous cellular machinery. The CRISPR and TALE systems use designer DNA binding factors to recruit repressor or activator domains (orange oval) to the lncRNA to affect transcriptional initiation. The effects of each strategy upon the process of transcription and presence of underlying DNA elements such as transcription factor binding sites are indicated. The possibility of generating stable transgenic animals to investigate phenotypes throughout development is also noted.

(*Marques et al., 2013*) and in these cases disruption of the lncRNA promoter could also cause unintended changes in gene expression. In the case of transcription terminators, to ensure effects are due to changes in RNA rather than DNA, inversions of the terminator sequence or a variety of different terminators can be used. In the experimental design it is also important to consider alternatively spliced transcripts and additional transcriptional start sites to ensure full abrogation of lncRNA expression.

Antisense oligonucleotides might provide an alternative technique for analysis of lncRNA function. They are thought to act by forming a DNA/RNA hybrid with the nascent RNA transcript, and triggering RNase H-dependent degradation of the RNA in the nucleus (*Figure 2*). This reduces the level of the RNA before the mature transcript

is produced, but the nature and extent of off-target effects are not fully understood and may be substantial (*Sahu et al., 2007*). Also, it is not possible to generate stable transgenic lines, which restricts analysis to cell lines or to systems where the oligonucleotides can be supplied by injection. Other approaches to disrupting lncRNA function use morpholino antisense oligos targeting e.g. splice sites (*Ulitsky et al., 2011*), or locked nucleic acid antisense oligonucleotides (*Sarma et al., 2010*).

Recent developments in rational design of DNA binding factors using transcription activator-like effector (TALE) proteins or the clustered regularly interspersed palindromic repeats (CRISPR) system have enabled recruitment of transcriptional activation (*Cheng et al., 2013*) or repression domains (*Cong et al., 2012*; *Gilbert et al., 2013*) to defined sites within the genome to modulate transcription, or to directly interfere with the passage of the RNA polymerase. These techniques could be used to modulate the rate of transcriptional initiation or elongation of the lncRNA (*Figure 2*), but care must be taken to control for direct effects of these factors on the transcription of neighbouring genes.

## Separating RNA- from DNA-sequence dependent effects

Deletion of a lncRNA genomic locus does not cleanly separate a role of the lncRNA per se from a role of other functional elements contained within the underlying DNA. Such elements might be irrelevant to the lncRNA's function, yet critical to the normal function of a neighbouring protein-coding gene. Eighteen mouse knockout lines were recently described in which genomic regions containing intergenic lncRNA loci (21.6 kb mean size, 4.8 kb–49.7 kb range) were deleted and replaced by a *lacZ* reporter cassette (*Sauvageau et al., 2013*). For 13 of these lines no overt phenotypes were reported. In contrast, strong phenotypes from 5 knockout lines were observed: $Peril^{-/-}$ or $Fendrr^{-/-}$ mice have reduced viability; $Mdgt^{-/-}$ and $linc\text{-}Pint^{-/-}$ mice show growth defects; and $linc\text{-}Brn1b^{-/-}$ mice exhibit abnormal cortical anatomy. The authors conclude that these developmental disorders generated by DNA deletions demonstrate the critical roles that lncRNAs play in vivo (*Sauvageau et al., 2013*).

While this may be the correct interpretation, the strong phenotypes observed in these lines may derive from the engineered deletion of *cis*-regulatory DNA elements lying within these large DNA deletions that are critical for the normal functions of proximal protein-coding genes. For instance *Fendrr* is 1.4 kb from *Foxf1*, and *Mdgt*

starts only 84 bp from the 5' exon of *Hoxd1* and terminates close to *Hoxd3* (*Figure 3*). Consistent with this notion, data from the ENCODE project indicate that the genomic region deleted in $Mdgt^{-/-}$ lines contains binding sites for several transcription factors and chromatin regulatory proteins (*Figure 3*). Whilst the authors detected no global change of neighbouring protein-coding gene expression as assessed by limited RNAseq of tissues, it is still possible that altered cell type or developmental stage specific expression of these genes escaped detection. LncRNAs are often transcribed in a highly restricted cell population and a global, high-throughput analysis of even the full embryo may not have been informative. Ultimately, the best evidence for RNA-dependent lncRNA function derives from loss-of-function, followed by complementation approaches, as for example described in *Grote et al. (2013)*.

This issue is also relevant for other lncRNAs transcribed from within *Hox* gene clusters. In the case of *Hotair* (*Rinn et al., 2007*), a several kb large deletion of the entire *Hotair* genomic DNA in vivo induces a subtle morphological phenotype in the spine, which was interpreted as a gain-of-function of *Hoxd* genes in *trans* (*Li et al., 2013*). However, *Hotair* is embedded in the *HoxC* gene cluster and topological modifications or re-arrangements in such a dense series of transcription units are likely to modify the expression of neighbouring genes. Further insights have been acquired by removing the entire *HoxC* locus, including both the lncRNA locus and flanking genes (*Suemori and Noguchi, 2000*; *Schorderet and Duboule, 2011*). Even when multiple alleles are available, as for *Hotair*, lncRNA function remains difficult to evaluate.

## Expression specificity and allelic series

Deletion of the mouse *Hotair* lncRNA also induced a subtle developmental phenotype in the wrist (*Li et al., 2013*). However, because murine *Hotair* transcripts were not detected in developing forelimb buds (*Schorderet and Duboule, 2011*) it remains possible that this phenotype develops from a lack of *Hotair* RNAs during subsequent stages of wrist development. This possibility could only be assessed by further analysis of the expression pattern of this lncRNA. The systematic introduction of a reporter cassette into lncRNAs (*Sauvageau et al., 2013*) can help solve this problem, provided the difference between the stability of the reporter staining and the half-life of the RNA is kept in mind, in particular for small and dynamic cell populations (*Zakany et al., 2001*).

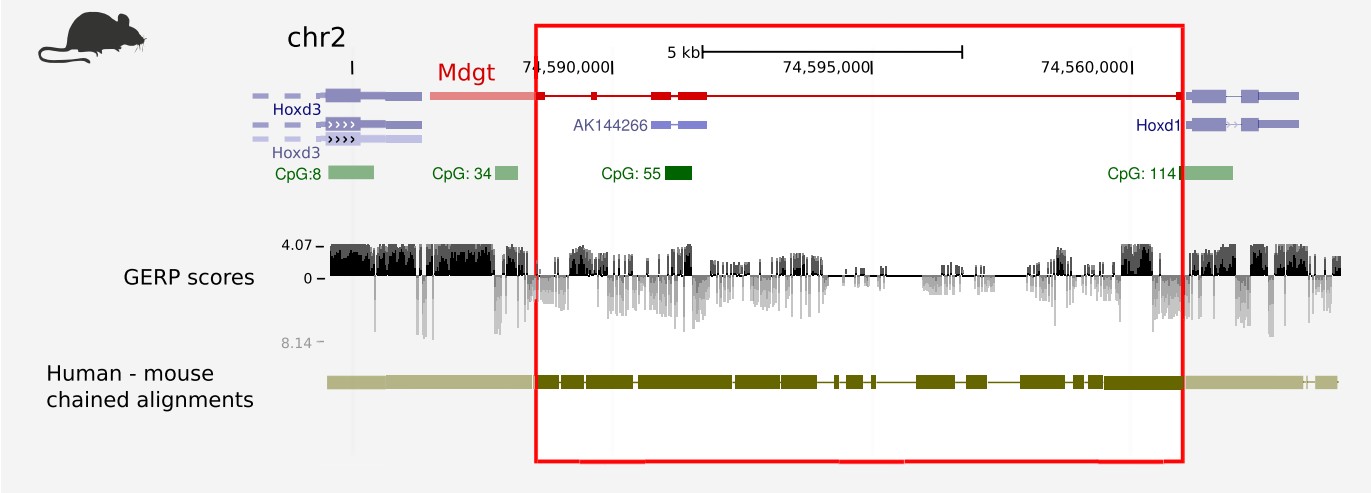

**Figure 3**. Human and mouse ENCODE data indicate that *Mdgt*⁻/⁻ lines contain deletions of conserved binding sites for transcription factors and chromatin regulatory proteins. The engineered deletion in mouse, and its equivalent sequence in human, are indicated by red rectangles, and spans 85% (12.4 kb of 14.7 kb) of intergenic sequence between mouse *Hoxd1* and *Hoxd3*. *Mdgt*, virtually shares its start site with *Hoxd1*, a gene expressed with exquisite specificities in only a few cell populations during early development (*Zakany et al., 2001*). Predicted transcription factor binding sites (TFBs) that are conserved in human, mouse and rat are shown against the human genome (*Consortium, 2012*; *Ernst and Kellis, 2012*). Numbers of experimentally-determined TFBs per genomic interval are shown in the histogram, and clusters of DNase 1 hypersensitivity sites, are also shown aligned against the human locus. Predicted CpG islands acquired from the UCSC Genome Browser are shown in green, and chained human-mouse alignments are shown in olive green. Evolutionary conservation (GERP) scores are indicated below the mouse locus.

As for protein-coding genes, an exhaustive description of functional traits associated with a particular lncRNA cannot be achieved by using a single mutant allele, hence allelic series are necessary. As indicated above, the nature of the alleles required to assess the function of a given

lncRNA depends upon its genomic location and its expression specificity during development and adulthood. This can be quite challenging, as exemplified by the bidirectional *Hotdog* and *Twin of hotdog* lncRNAs: even though these RNAs are located hundreds of kb distant from the *HoxD* gene cluster in the middle of a gene desert, their shared start site physically interacts with *Hoxd* genes as part of a general regulatory structure. In this case, a *cis*-effect could in principle be evaluated by separating the lncRNA loci from the *HoxD* cluster *via* a large inversion with a breakpoint in-between. It turns out, however, that this inversion globally disrupts the regulation of *HoxD* by displacing long-range acting enhancers along with the lncRNA loci, making interpretation difficult (*Delpretti et al., 2013*).

## Discrepancies between different strategies

The lncRNA *Fendrr* has been studied using two independent strategies: genetic deletion (*Sauvageau et al., 2013*) and transcriptional terminator insertion (*Grote et al., 2013*). Whilst both studies describe a lethal phenotype, highlighting the potential importance of this lncRNA in development, the outcomes differ. Genetic deletion results in lung maturation and mesenchymal differentiation defects (*Sauvageau et al., 2013*), whilst terminator insertion leads to heart and body wall defects and to effects on the expression of the neighbouring *Foxf1* gene (*Grote et al., 2013*). Importantly, the defects caused by terminator insertion were rescued by a transgene containing a single wild type copy of the *Fendrr* lncRNA locus (without its functional *Foxf1* neighbour); this strongly implicates deletion of the RNA product, rather than its genomic DNA, as causing the observed phenotypes (*Grote et al., 2013*). Transgene rescue experiments are thus crucial for establishing RNA-dependent lncRNA function. An earlier successful illustration of this principle was the rescue of developmental defects in zebrafish by co-injection of spliced RNA for each of two lncRNAs, *cyrano* and *megamind*, whose precursor RNAs had been knocked down using morpholino antisense oligos (*Ulitsky et al., 2011*). However, regulatory sequences necessary for the transcription of the lncRNA itself should ideally be included in the rescue construct so as to maintain physiological levels of expression. This, added to the length of lncRNAs that can sometimes reach several hundred kb, may represent a challenge for a transgenic approach.

Substantial differences have also been observed between RNAi-mediated knockdown and transcriptional terminator insertion at the *Evf-2* lncRNA

locus (*Feng et al., 2006*; *Bond et al., 2009*; *Berghoff et al., 2013*; *Kohtz, 2014*). This lncRNA is transcribed across an enhancer element between the *Dlx5* and *Dlx6* genes, and initial studies in cell culture using RNAi suggested a model whereby *Evf-2* was important for activation of *Dlx5/6* (*Feng et al., 2006*). However, transcriptional terminator insertion in mice has shown the opposite effect on expression of *Dlx5/6* (*Bond et al., 2009*) and causes specific changes in DNA methylation at the enhancer. Importantly these changes can be rescued by *Evf-2* expression from a separate transgene, implying that they are dependent on the lncRNA itself (*Berghoff et al., 2013*).

Similarly to this example, knockdown of *linc*RNA-*p21* by RNAi originally suggested a *trans*-acting mechanism, in which the lncRNA was involved in recruiting protein complexes to chromatin (*Huarte et al., 2010*). Nevertheless, subsequent studies where the promoter of the lncRNA was deleted or its transcription was blocked by antisense oligonucleotides have highlighted a different role, as this lncRNA regulates the adjacent *p21* gene in *cis*, without having *trans*-acting effects (*Dimitrova et al., 2014*). Whilst both studies analysed by RNAseq the effect of lncRNA depletion on global gene expression in mouse embryonic fibroblasts, the two sets of differentially expressed genes did not overlap significantly. When analysing lncRNA function, it is thus important to consider multiple loss-of-function strategies that address multiple mechanisms of action.

The potential confounding effects of techniques used to separate DNA- from RNA-dependent function are further exemplified by studies of the *Drosophila bxd* lncRNA, which is expressed from within the HOX cluster, adjacent to the *Ultrabithorax* (*Ubx*) gene. Its expression is highly specific and occurs in the same broad region of the embryo as the *Ubx* gene, although notably never within the same cell (*Petruk et al., 2006*). Studies of *bxd* loss-of-function using different techniques have yielded conflicting interpretations. It has long been known that small deletions within this lncRNA cause dramatic effects on expression of the neighbouring *Ubx* gene (*Lewis, 1978*), resulting in homoeotic transformations. Indeed, certain allelic combinations are able to generate a four-winged fly. More recent studies of the same deletions suggest that the act of transcription of this lncRNA represses *Ubx* in *cis* by altering protein binding to the *Ubx* promoter (*Petruk et al., 2006*). In contrast, it was reported that inversion of the *bxd* promoter, driving transcription in the wrong direction whilst maintaining genomic composition, results in very minor effects on *Ubx*

expression, and then only later in development (*Pease et al., 2013*). Also, a deletion removing the promoter induced a *Cdx*-like gain of function of *Ubx* (*Sipos et al., 2007*). Clearly, correct interpretation of such loss-of-function experiments, at such complex loci, requires careful consideration of potentially confounding factors.

Contrasting results of different experiments may also arise because of a lncRNA's involvement in different mechanisms in different cellular contexts. For example, in embryonic cells, transcription of *Airn* silences the adjacent *Igf2r* gene (*Latos et al., 2012*), whereas in extraembryonic tissues it acts more distally by recruiting the histone methyltransferase G9a to imprinted genes (*Nagano et al., 2008*).

## The end of the beginning: a maturing lncRNA field

The study of lncRNAs is still in its infancy, and the biochemical and genetic techniques used to address the true significance and mechanisms of action of this class of RNA have only recently been developed or adapted from those used for investigating protein-coding genes. Such techniques must therefore be used with caution and with appropriate controls (*Brockdorff, 2013*; *Riley and Steitz, 2013*). From the examples described above, it is apparent that the optimal strategy with which to study a lncRNA's loss of function depends both on the mechanism by which it acts, in particular in a *cis* or *trans* configuration, and the regulatory sequences present within its locus. We suggest that early lessons learnt from paradigm repressor lncRNAs, such as *Xist*, and imprinted lncRNAs such as *Airn* or *Kcnq1ot1*, should guide the design of experiments on more recently identified lncRNAs. We have attempted to distil these lessons into the proposed considerations in *Box 1*. Introduction of the multiple alleles that will be necessary to adequately dissect lncRNA in vivo function will be greatly aided by recent advances in genome engineering using designer site-specific nucleases such as CRISPR/Cas9 and TALENs. The introduction of fast acute loss-of-function systems for lncRNAs, for example those that insert a sequence-specific ribonuclease site whose nuclease is under drug inducible control, would also greatly facilitate lncRNA investigation.

The *trans* function of a lncRNA may be investigated using locus deletion, promoter deletion, inversions, transcriptional termination or RNAi. Where possible, these strategies should be combined with genetic rescue experiments, where the lncRNA is expressed from an independent transgene inserted at a location distinct from the lncRNA locus. This strategy separates RNA-dependent effects from those arising from the manipulation of the underlying DNA. Rescue experiments using expression of the lncRNA from an independent transgene are only possible for *trans*-acting lncRNAs where the RNA moiety itself and not the act of transcription is critical for function.

The *cis* function of a lncRNA may be investigated using a combination of several alleles, such as insertion of transcriptional terminators, promoter deletions and inversions. Several alleles are likely to be required to separate lncRNA-dependent from other effects and, as controls, to reveal artefacts of genetic engineering. Engineered inversions can also be used to separate the lncRNA locus from its potential neighbouring target gene to investigate its roles in *cis*. Use of site-specific recombinases, such as the phiC31/attP system (*Bateman et al., 2006*; *Zhu et al., 2014*) as 'landing sites' or for recombination mediated cassette exchange, will greatly enhance our ability to generate such allelic series. For example, the lncRNA locus may be deleted and replaced by a recombinase 'landing site' into which different constructs can be introduced to investigate phenotype rescue.

In summary, if lncRNA biologists are to resolve the true in vivo functions of these numerous and enigmatic transcripts, then the strengths and weaknesses of available techniques will need to be acknowledged. Resolution will no doubt derive from the careful and comprehensive genetic dissection of individual loci using multiple alleles. The field of lncRNA biology would benefit greatly from the development of additional approaches that are effective in distinguishing effects mediated by lncRNAs as molecular species from their effect on gene regulatory elements with which lncRNA loci are interleaved across the mammalian genome.

### Acknowledgements

AB & CPP: the European Research Council (DARCGENs, project number 249869) and the Medical Research Council. AA: Max Planck Institute. DD: the European Research Council (Systems*Hox*.ch) and the Swiss National Research Foundation. APB: the Wellcome Trust. DPB: Austrian Academy of Sciences and the Austrian Science Fund FWF F4302-B09. AE: the European Molecular Biology Laboratory. AFS: The Wellcome Trust and MRC. DRH: the Medical Research Council. EAM: Cancer Research UK. TRG: NHGRI U54 HG007004-2.

**Andrew R Bassett** is in the MRC Functional Genomics Unit, Department of Physiology, Anatomy and Genetics, University of Oxford, Oxford, United Kingdom.
*Present address:* Sir William Dunn School of Pathology, University of Oxford, Oxford, United Kingdom

**Asifa Akhtar** is in the Department of Chromatin Regulation, Max-Planck-Institut für Immunbiologie und Epigenetik, Freiburg im Breisgau, Germany

**Denise P Barlow** is in the CeMM, Research Center for Molecular Medicine of the Austrian Academy of Sciences, Vienna, Austria

**Adrian P Bird** is in the Wellcome Trust Centre for Cell Biology, University of Edinburgh, Edinburgh, United Kingdom

**Neil Brockdorff** is in the Department of Biochemistry, University of Oxford, Oxford, United Kingdom

**Denis Duboule** is in the School of Life Sciences, Ecole Polytechnique Fédérale Lausanne, Lausanne, Switzerland; Department of Genetics and Evolution, Université de Genève, Geneva, Switzerland

**Anne Ephrussi** is in the Developmental Biology Unit, European Molecular Biology Laboratory, Heidelberg, Germany
http://orcid.org/0000-0002-5061-4620

**Anne C Ferguson-Smith** is in the Department of Genetics, University of Cambridge, Cambridge, United Kingdom

**Thomas R Gingeras** is in the Functional Genomics Group, Cold Spring Harbor Laboratory, Cold Spring Harbor, United States

**Wilfried Haerty** is in the MRC Functional Genomics Unit, Department of Physiology, Anatomy and Genetics, University of Oxford, Oxford, United Kingdom

**Douglas R Higgs** is in the MRC Molecular Haematology Unit, Weatherall Institute of Molecular Medicine, Oxford, United Kingdom

**Eric A Miska** is in the Wellcome Trust Cancer Research UK Gurdon Institute, University of Cambridge, Cambridge, United Kingdom; Department of Genetics, University of Cambridge, Cambridge, United Kingdom

**Chris P Ponting** is in the MRC Functional Genomics Unit, Department of Physiology, Anatomy and Genetics, University of Oxford, Oxford, United Kingdom; Wellcome Trust Sanger Institute, Cambridge, United Kingdom
http://orcid.org/0000-0003-0202-7816

**Author contributions**
ARB, CPP, Conception and design, Drafting or revising the article; AA, DPB, APB, NB, DD, AE, ACF-S, TRG, WH, DRH, EAM, Drafting or revising the article

*Competing interests:* CPP: Senior Editor, *eLife*. AA: Reviewing Editor, *eLife*. ACF-S: Reviewing Editor, *eLife*. TRG: Reviewing Editor, *eLife*. The other authors declare that no competing interests exist.

## Funding

| Funder | Grant reference number | Author |
| --- | --- | --- |
| European Research Council | DARCGENS 249869 | Andrew R Bassett, Chris P Ponting |
| Medical Research Council | | Anne C Ferguson-Smith, Wilfried Haerty, Douglas R Higgs, Chris P Ponting |
| Max-Planck-Gesellschaft | | Asifa Akhtar |
| European Research Council | SystemsHOX.ch | Denis Duboule |
| Swiss National Science Foundation | | Denis Duboule |
| Wellcome Trust | | Adrian P Bird, Anne C Ferguson-Smith |
| Austrian Academy of Sciences | | Denise P Barlow |
| Austrian Science Fund | FWF F4302-B09 | Denise P Barlow |
| European Molecular Biology Laboratory | | Anne Ephrussi |
| Cancer Research UK | | Eric A Miska |
| National Human Genome Research Institute | U54 HG007004-2 | Thomas R Gingeras |

The funders had no role in study design, data collection and interpretation, or the decision to submit the work for publication.

## Additional files

### Major dataset

The following previously published dataset was used:

| Author(s) | Year | Dataset title | Dataset ID and/or URL | Database, license, and accessibility information |
| --- | --- | --- | --- | --- |
| Encode Project Consortium | 2012 | ENCODE - Encyclopedia of DNA elements | https://genome.ucsc.edu/ENCODE/ | Freely available at ENCODE Data Coordination Center (DCC). |

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
