## [Decision Letter]

Thank you for sending your work entitled “Considerations when investigating lncRNA function in vivo” for consideration at *eLife*. Your article has been favorably peer reviewed by Detlef Weigel and two outside reviewers.

There has been considerable recent excitement about the role of long noncoding RNAs (lncRNAs) in gene regulation. A major difficulty in studying lncRNA function genetically is that they can overlap with other elements of the genome that have proven (in the case of antisense RNA) or potential separate functions (such as insulators, enhancers, etc). Therefore, simple knockout experiments, using either insertion of large foreign sequences or sequence deletions may produce misleading results.

The article provides several examples from the literature that clearly illustrate how studies of the same lncRNA locus can come to different conclusions. The article is timely given the apparent lack of defined “gold standards” for this kind of work. A variety of approaches for the functional analysis of lncRNA loci is discussed, highlighting the pitfalls that can be encountered when trying to distinguish the contributions of genomic DNA from the act of transcription or the RNA product itself to the observed phenotypes. The difficulties associated with distinguishing cis versus trans effects of lncRNAs are discussed and rigorous and standardized approaches to evaluating lncRNA function in vivo are proposed. The article will thus serve as an important guide to those performing genetic analyses of lncRNA loci.

The discussion is overall very balanced and the reviewers had only a few suggestions:

1) Include either at the beginning or at the end a series of bullet points, with an introductory sentence that lncRNA functional evidence should include at least x out of y of these types of evidence.

2) Mention the need for appropriate controls, when genetic manipulations, e.g., in ES or similar cells, are followed by selection of clones derived from single cells. ES and other cultured cells (e.g., derived from tumors) are genetically not perfectly stable, and one needs to ensure that phenotypes are not due to second-site mutations. Multiple clones or transgenic rescue experiments should be the norm (obviously this applies not only to lncRNA analyses).

3) It would be best to distinguish true “loss of function”, in which transcription is completely abolished, from “reduction of function”, where a truncated transcript is produced or RNAi is used. To this end, Figure 2 should be amended.

4) A brief paragraph that mentions flanking cell biological (e.g., FISH) and biochemical (e.g., RNP analysis) approaches for evaluating lncRNA function would be worthwhile.

---

## [Author Response]

*1) Include either at the beginning or at the end a series of bullet points, with an introductory sentence that lncRNA functional evidence should include at least x out of y of these types of evidence*.

The intention of Box 1 “Considerations when interpreting phenotypes resulting from lncRNA mutation” was to provide such suggestions. However, we believe that each lncRNA has to be considered on its own merits, depending on its mechanism of action and it is therefore not possible to define a set of general guidelines that would be applicable to all cases. We have therefore deliberately given a set of “considerations” rather than “guidelines”.

*2) Mention the need for appropriate controls, when genetic manipulations, e.g., in ES or similar cells, are followed by selection of clones derived from single cells. ES and other cultured cells (e.g., derived from tumors) are genetically not perfectly stable, and one needs to ensure that phenotypes are not due to second-site mutations. Multiple clones or transgenic rescue experiments should be the norm (obviously this applies not only to lncRNA analyses)*.

We agree with the reviewers that this is an important point, since genetic background will have a large effect, especially when the phenotypes are subtle, and that this can vary substantially between different clones or lines. We have added this concept to the text.

*3) It would be best to distinguish true “loss of function”, in which transcription is completely abolished, from “reduction of function”, where a truncated transcript is produced or RNAi is used. To this end,*
Figure 2
*should be amended*.

We have amended Figure 2 and the text to make this clear.

*4) A brief paragraph that mentions flanking cell biological (e.g., FISH) and biochemical (e.g., RNP analysis) approaches for evaluating lncRNA function would be worthwhile*.

We have added a paragraph describing some of the complimentary techniques that can be used to evaluate lncRNA mechanism, and how these may be used to guide mutant design.